# Genetic Background Matters: Population-Based Studies in Model Organisms for Translational Research

**DOI:** 10.3390/ijms23147570

**Published:** 2022-07-08

**Authors:** Valeria Olguín, Anyelo Durán, Macarena Las Heras, Juan Carlos Rubilar, Francisco A. Cubillos, Patricio Olguín, Andrés D. Klein

**Affiliations:** 1Centro de Genética y Genómica, Facultad de Medicina, Clínica Alemana Universidad del Desarrollo, Santiago 7610658, Chile; valeria.olguin.araneda@gmail.com (V.O.); anduranm@udd.cl (A.D.); mlasherasp@udd.cl (M.L.H.); jrubilare@udd.cl (J.C.R.); 2Departamento de Biología, Santiago, Facultad de Química y Biología, Universidad de Santiago de Chile, Santiago 9170022, Chile; francisco.cubillos.r@usach.cl; 3Millennium Institute for Integrative Biology (iBio), Santiago 7500565, Chile; 4Program in Human Genetics, Institute of Biomedical Sciences, Biomedical Neurosciences Institute, Department of Neuroscience, Facultad de Medicina, Universidad de Chile, Santiago 8380453, Chile; patricioolguin@uchile.cl

**Keywords:** systems genetics, mouse, *Drosophila*, *Saccharomyces cerevisiae*, translational research, genetic background, precision medicine, gene mapping

## Abstract

We are all similar but a bit different. These differences are partially due to variations in our genomes and are related to the heterogeneity of symptoms and responses to treatments that patients exhibit. Most animal studies are performed in one single strain with one manipulation. However, due to the lack of variability, therapies are not always reproducible when treatments are translated to humans. Panels of already sequenced organisms are valuable tools for mimicking human phenotypic heterogeneities and gene mapping. This review summarizes the current knowledge of mouse, fly, and yeast panels with insightful applications for translational research.

## 1. Precision Medicine in Humans

Precision medicine characterizes diseases at a higher resolution by genomic and other technologies, providing more accurate targeting of patient subsets with tailored therapies [1]. To make this possible, large genotyped cohorts with deep clinical annotations are required to map loci responsible for the phenotypic variability. Common approaches to gene mapping include genome-wide association studies (GWAS) and linkage analysis in families of patients with variable disease severity [1]. These studies are time-consuming and expensive due to recruiting and genotyping costs. Furthermore, it is virtually impossible with rare diseases to find large cohorts in order to assure statistical significance for the genomic mapping.

Furthermore, families presenting enough informative individuals with variable symptoms are challenging to identify [2]. Strategies using model organisms with various genetic backgrounds are valuable resources for overcoming these obstacles. In this review, we describe many panels of organisms and examples of how modeling diseases on them can accelerate the pace of discoveries toward translational research in humans.

## 2. Rodents as Model Organisms in Genetic Research: Advantages and Limitations

The advantages of using mouse models in biomedicine have been discussed extensively [3]. Some benefits are the following: (i) the availability of genetic tools for creating disease models by transgenic, knockout, and knock-in technologies [4,5,6] (https://www.jax.org/research-and-faculty/resources/mouse-mutant-resource/available-models (accessed on 22 May 2022)); (ii) inbred mouse strains are nearly isogenic, enabling to study how the same genetic mutation modifies a phenotype of interest in different genetic backgrounds [7,8,9,10,11]; (iii) mouse tissues are available for omics studies which can be challenging to obtain from humans [12]. Some limitations include different evolutive pressures for mice and humans; therefore, some systems, such as the immune system, do not function similarly in both species [13].

### 2.1. Hybrid Mouse Diversity Panel

Currently available resources in rodents to find modifiers genes by association studies can be defined in two categories: (i) reference panels, consisting of inbred strains such as the Hybrid Mouse Diversity Panel (HMDP) and the Collaborative Cross (CC); (ii) populations derived from pseudo-random breeding of inbred strains, such as the Diversity Outbred (DO) and Heterogeneous Stock (HS) (Figure 1).

HMDP is a large panel of approximately 100 commercially available (https://www.jax.org (accessed on 22 May 2022)) and fully sequenced (www.sanger.ac.uk/science/data/mouse-genomes-project (accessed on 22 May 2022)) inbred strains: ~30 classical inbred strains and ~70 recombinant inbred (RI) strains derived mainly from crosses between C57BL/6J and DBA mice and A/J and C57BL/6J mice [14].

Advantages of using the HMDP panel are the following: (i) their genomes are known (http://mouse.cs.ucla.edu/mouseHapMap/ (accessed on 22 May 2022)); thus, it is unnecessary to spend funds performing this step; (ii) HMDP possesses ~4 million common single-nucleotide variants (SNVs), which is similar to the number present in humans [15]; (iii) high-resolution association mapping [14], which is at least an order of magnitude higher than in linkage analysis; (iv) it is possible to integrate gene mapping with other omics (transcriptomics, proteomics, and metabolomics data) [12]; (v) commercially available (from The Jackson Laboratory, Harlan, and others); (vi) sufficient bioinformatics tools for data mining of complex mouse and human disease traits, such as the Systems Genetics Resource (SGR) (http://systems.genetics.ucla.edu (accessed on 22 May 2022)); (vii) servers to perform association mapping and statistical power simulation, which are also available in R to run them in house [16].

The HMDP also has limitations. For example, extensive linkage disequilibrium (LD) blocks are observed, both within and between chromosomes, probably as a result of the selection of allelic combinations conceding higher fitness during the inbreeding [17]. Consequently, regions in LD can lead to false-positive associations in GWAS analyses. Although the HMDP has a high mapping resolution, the statistical power to detect the effect of loci is small (estimated at 50% to variants explaining 10% of the trait variance) [14]. Since most loci contributing to a complex trait have an effect size below 5% [18], variants with subtle effects cannot always be detected by the HMDP. Power can be enhanced by including additional inbred and RI strains and performing meta-analyses from other panels such as the CC or traditional crosses [19].

An exciting application of the use of mouse panels in translational research comes from crossing the classical Alzheimer’s disease (AD) mouse model (5XFAD) bearing mutations in APP and PSEN1 with 28 different strains of the BXD panel (AD-BXD). The F1 represents isogenic lines that were studied in a controlled environment. The AD-BXD panel mimicked several signs of the AD patients, including phenotypic variation in disease onset and severity. As in humans, the Apoe allele significantly affected spatial memory and other behavioral tests in the AD-BXD panel. Furthermore, hippocampal gene expression in the severe and mild lines agrees with transcriptomic changes observed in patients [20].

### 2.2. The Collaborative Cross (CC) Panel

The CC is a large panel of RI mouse strains obtained through systematically outcrossing eight founder strains, followed by randomized breeding [21]. The founder strains of the CC include five of the widely used classical inbred laboratory strains (A/J, C57BL/6J, NOD/ShiLtJ, 129S1/SvImJ, and NZO/HILtJ), as well as three wild-derived strains descendent of three *M musculus* subspecies (WSB, Castaneous, and PWK) (Figure 1). These eight strains have been fully sequenced and carry ~45 million SNVs, four times more than those of classical laboratory mouse strains [22].

The genomes of the CC panel are known (http://csbio.unc.edu/CCstatus/CCGenomes (accessed on 22 May 2022)), which is helpful for genetic association studies. Haplotypes can be easily visualized or reconstructed as a mosaic of the genomes of the founders [23]. Parental strains capture approximately 90% of the genetic diversity seen in the *Mus musculus* species [24]. This high genetic diversity significantly reduces false candidate loci. Additionally, randomized breeding substantially increases mapping resolution by reducing population structure effects [25]. CC strains have been used to map quantitative trait loci (QTLs) to less than 5 Mb intervals [26]. Online tools are available to perform GWAS and linkage analyses [27]. Several aspects of human genetics and behavioral factors can be modeled in this system, including the heterogeneities observed in neurodevelopmental disorders such as autistic spectrum disorders (ASDs) [28]. The CC panel allowed the discovery of novel candidate severity modifiers of ASD, e.g., *Bai3,* considered a potential target for pharmacological intervention [28].

Some considerations associated with using the CC panel are the following: (i) unique outlier phenotypes can arise in large studies, probably due to the complex genetic regulatory networks involving multiple loci with epistatic interactions [29]; in such cases, the preferred approach for identifying causal genes is traditional F2 analysis or backcrosses [30]; (ii) because identifying loci could be time-consuming, it is suggested to perform a pilot study and expand as necessary [29]; (iii) creating a panel like the CC can generate breeding complications and infertility, mainly caused by genomic incompatibility introduced by the wild-derived strains. For that reason, the initial CC project aimed to produce 1000 strains but finished with only ~100 and inspired the creation of the Diversity Outbred (DO) population.

CC lines have been used for genetic association studies of many complex traits. QTL mapping for 15 metabolism- and exercise-related traits revealed five significant loci for body weight, some of which overlapped with previous human studies [31]. Gene mapping of rotarod (exercise) performance and body weight identified 45 loci, many of them related to neurological disorders and obesity in humans, suggesting a link between physical activity and neurodegeneration [32]. A study of glucose tolerance response in the CC panel identified, only in female mice, a genomic region comprising 51 genes. This study highlighted sex differences in glucose response which should be considered in human studies [33]. The CC panel is also a valuable and reliable resource for studying host–pathogen interactions [29]. For example, to map genetic modifiers affecting the severity of *Pseudomonas aeruginosa* lung infections, 39 CC lines were inoculated with this pathogen. The phenotypic variability was enormous, ranging from complete resistance to lethality. It is particularly relevant to study the resistant lines since they have the biological secrets to design novel therapies for the susceptible. Genomic mapping and functional validation identified dihydropyrimidine dehydrogenase (*Dpyd*) and sphingosine-1-phosphate receptor 1 (*S1pr1*) as modifier genes. In a cohort of patients with cystic fibrosis, two SNVs in the *S1PR1* gene are associated with *Pseudomonas aeruginosa* infection [34], again indicating the translational relevance of multigenetic background studies in animal organisms.

### 2.3. Heterogeneous Stock and Diversity Outbred Populations

Both HS and DO are high-diversity outbred mice populations. The HS was established by breeding eight inbred strains and then outbreeding them in either a circular strategy or using random crosses (Figure 1) to minimize inbreeding [35]. After 50 or more generations, the HS-generated mice were a genetic mosaic of the founders’ haplotypes [36,37]. On the other hand, the DO was established from partially inbred CC lines and is maintained indefinitely through pseudorandomized fashion non-sibling mating [38] (Figure 1). Since the DO is derived from the same eight founders as the CC, it presents the same allelic diversity as the CC strains. It can be used as a complementary tool in genetic association studies [39].

There are several advantages of using HS or DO mice compared to classical inbred mice. The outbred randomized mating increases the number of additional recombination sites compared to those of classically inbred mice; thus, each HS or DO mouse has a unique genome, which is a mosaic of the original eight founder lines, resembling human heterozygosity and allows high-resolution genetic mapping [39]. HS and DO mice have been used to finely map to intervals of 2.7 Mb [40] and less than 2 Mb [39], respectively. In addition, outbred animals are more vigorous and less prone to both early and late recessive allelic effects [41]. This genetic variability within both HS and DO populations results in a high degree of phenotypic variability; thus, outbred models enable the fine mapping of many phenotypic traits. Since the founders of CC and DO lines include wild-derived strains, unique behaviors can be observed compared to classical laboratory strains and represent a valuable tool for genetic behavior association studies [22]. A repository of DO QTL studies can be shared between laboratories (https://dodb.jax.org (accessed on 22 May 2022)). Lastly, the founders of the HS and DO populations have been sequenced [42], reducing time and expense in locating the sequences.

Alternatively, some considerations must be made in the case of HS and DO mice. Since each outbred animal is genetically and phenotypically distinct, each HS and DO mouse requires genotyping and haplotype reconstruction to perform each QTL analysis [38]. High-resolution mapping can be achieved with these panels, but analyzing many animals is necessary for sufficient statistical power, which is not always possible [43]. Candidate modifiers of wild behaviors can be identified with outbred mice. However, it is challenging to validate in these panels because each animal has a unique genotype, in contrast to inbred lines [44].

An interesting translational study using the DO panel identified a diagnostic biomarker for human tuberculosis (TB). By applying machine learning algorithms to multidimensional data, the authors discovered CXCL1 as a putative biomarker of TB in the serum of mice. The biomarker was further validated in samples derived from human patients, discriminating active TB from latent infection and non-TB lung disease [45]. This study highlights the relevance of using population-based strategies to accelerate human biomarker discovery, validation, and testing.

## 3. *Drosophila melanogaster* as a Model Organism in Genetic Research: Advantages and Limitations

In addition to mouse models, *Drosophila melanogaster* has attracted many scientists. Flies are small, easy to manipulate in the laboratory, and cheap to maintain. They have a short life span (2 week generation interval) and produce many offspring. Flies show complex behaviors, including sleep, aggression, addiction, and social behavior [46]. Notably, about 70% of human disease-associated genes have a *Drosophila* ortholog [47]; its genome is fully sequenced and well annotated. It can be genetically modified using chemical and insertional mutagenesis, gene-specific mutations, or editions using CRISPR [47,48]. These characteristics support its use as a model system to study human diseases. As expected, the use of *Drosophila* for human research has limitations; for instance, the fly does not possess hemoglobin [49] and, thus, cannot be used for studying human pathologies related to this system.

### 3.1. Drosophila melanogaster Genetic Reference Panel (DGRP)

The DGRP is a collection of 205 inbred *Drosophila melanogaster* strains derived from a single natural population. Inseminated females were collected from the farmer’s market in Raleigh, NC (USA), and their offspring were subjected to 20 generations of complete sibling mating [50] (Figure 2). The DGRP is a public resource available at the Bloomington Drosophila Stock Center (http://fly.bio.indiana.edu (accessed on 22 May 2022)) built for genomic association analyses. Currently, their genomes are available, and each line has minimal genetic variation [50]. Repeated measurements within each line are possible, enabling accuracy to increase the statistical power in GWA analyses. Since the DGRP is a publicly available resource, it allows different laboratories to correlate phenotypes on the same genotype and understand the pleiotropic effects of DNA variants and genes on multiple quantitative traits. Unlike the human genome, the fly genome has a structure with low LD between closely linked polymorphisms [51], which is favorable for accurate association mapping; thus, significant associated SNVs are likely causal or very near to a causal variant [52]. Lastly, experimentation in *Drosophila* has fewer ethical concerns compared to rodent models.

As with all study models, there are some limitations in DGRP that should be considered. Firstly, genetic variation between the lines is a snapshot of the population from which they were derived; therefore, DGRP does not represent all the possible variations of the species. Secondly, the 205 lines usually provide enough statistical power to detect common variants with moderate to large effects [53,54], but the statistical power is still limited for rare variants (minor allele frequency (MAF) < 0.05) [51].

### 3.2. DGRP for Mapping Physiological and Pathophysiological Traits

The DGRP has been used for GWA mapping of many different quantitative physiological traits, including food intake and sleep behavior [55,56]. Food intake is essential to animal fitness, and 25 modifiers with human orthologs were found [55]. Interestingly, diversity in mitochondrial haplotypes can directly mediate phenotypic variation in food intake [57]. Sleep has been increasingly explored in recent years with this model [56]. Flies resemble mammalian sleep and have become an important model species for identifying sleep regulation mechanisms. Analogous to human sleep studies, a DGRP GWAS highlighted signals in the EGFR, Wnt, Hippo, and MAPK signaling pathways, suggesting that genes affecting variation in this trait are conserved [58]. DGRP studies revealed the genetic architecture of nutrient stores (glucose, glycogen, glycerol, protein, triglycerides, and wet weight) [59], developmental plasticity [60], and circadian cycle [61].

The DGRP has been used to identify candidate modifiers of retinal degeneration [62] and neurodegeneration in a Parkinson’s disease (PD) model [63]. PD is a highly variable neurodegenerative disorder where variable manifestations range from cognitive disturbances, motor alterations, and sleep and speech abnormalities to cellular pathological changes such as the formation of Lewy body inclusions and neuronal death [64]. The leucine-rich repeat kinase 2 gene G2019S mutation (LRRK2 G2019S) penetrance is incomplete and varies among ethnic populations. In the Ashkenazy Jewish population, the low penetrance (26%) of the G2019S mutant phenotype suggests that other factors, such as the genetic background, the environment, and their interaction, act as modifiers of the variable phenotype [65,66]. In this regard, it has been reported that introducing the LRRK2 G2019S mutation in the DGRP results in considerable variability in the locomotor phenotype among backgrounds [63]. Gene mapping revealed 177 candidate modifier genes enriched in pathways involved in the neuronal outgrowth. The study suggests a link among LRRK2, neurite regulation, and neuronal degeneration in PD [63].

### 3.3. Lines Derived from DGRP and DSRP

A limitation of the DGRP is its low statistical power [51], which motivated the development of DGRP-derived advanced intercross populations (AIPs). These correspond to lines generated by crossing parentals DGRP for many generations, which were then remapped [67]. By successive crossings of a subset of parentals lines, it is possible to increase the recombination rate and, consequently, the statistical power compared to the DGRP [52]. Furthermore, the extreme QTL mapping strategy in AIPs can be used to resolve the statistical limitations of the DGRP for rare variants (MAF < 0.05). Extreme QTL mapping refers to selecting individuals from the extremes of the phenotypic distribution for a trait (resembling a case–control study). Flies are pooled and sequenced, which is cheaper than sequencing all individuals of the initial population. This allows identifying alleles that segregate differentially among the distribution extremes (causal variant or in LD with it) [68,69]. The discovery of rare variants in DGRP will occur at higher frequencies in the AIPs after an extreme QTL mapping strategy.

A less applied strategy to increase the mapping power is to use DGRP and another panel for cross-validation, such as the Drosophila Synthetic Population Resource (DSPR). This collection of 1700 inbred lines is derived from 15 isogenic founder lines created from geographically distinct *Drosophila* populations [70]. However, some studies in both AIPs and DSPR lack overlap with candidate genes found in DGRP, probably due to the different genetic architecture or genetic variants between the panels.

## 4. *Saccharomyces cerevisiae* as a Model Organism in Genetic Research: Advantages and Limitations

*Saccharomyces cerevisiae*, the budding yeast, has gained prominence as a model organism in quantitative genetics because it has several experimental and biologically advantageous features. For example, it has a small and compact genome of approximately 12 million bp in haploids (about one two-hundredth of the human genome). It contains fewer introns and a lower proportion of intergenic sequences than higher eukaryotes [71]. Furthermore, it is easy to cultivate and maintain in large population size in the laboratory. In addition, two-thirds of all yeast genes share at least one domain of significant homology with human genes, and about 30% of known genes involved in human diseases have yeast orthologs [72].

One of the main advantages of yeast for quantitative genetics studies is its large genetic map. *S. cerevisiae* exhibits high meiotic recombination rates, with an average of about 90 crossovers per meiosis, allowing precise quantitative phenotyping [71,73,74]. The homologous recombination in yeast is highly efficient, facilitating the deletion of sequences or genes in vivo [72,75]. This efficient recombination permitted the generation of the first complete deletion mutant strain collection using gene replacement with the G418 resistance gene (KanMX) cassette in the reference *S. cerevisiae* strain [76]. Since then, similar panels have been available in different genetic backgrounds, demonstrating the high degree of genetic background dependencies for different phenotypes [77,78]. Yeasts have less genetic complexity than flies and rodents. Thus, it is easier to study the effect of a single gene because of the reduced genetic redundancy [79].

### 4.1. Analysis of Segregating Populations from Pairwise Crosses

QTL mapping in yeast has been the primary approach to uncovering genetic variants responsible for phenotypic differences between genetic backgrounds. Identifying QTLs has been achieved by analyzing segregating populations from pairwise crosses, mainly through linkage or bulk segregant analysis (BSA) [80,81]. Linkage mapping in yeast involves mating two or more haploid parental strains that show phenotypic variation and then phenotyping and genotyping a panel of recombinant offspring obtained from these crosses. Recombination breaks allow causal loci to segregate with the phenotype of interest, and QTLs are identified using statistical tests [80,82]. The BSA also involves crossing two or more parental strains and subsequent phenotyping of their recombinant offspring [83]. However, the BSA method uses selective genotyping of subsets of segregants, commonly the extremes of the phenotypic distribution [84]. Typically, segregants undergo selective environmental pressure, where large pools are constructed. One expresses the trait of interest (selected pool), and others are not selected (control pool) or exhibit the opposite phenotype. After genotyping each marker, genetic regions of allelic enrichment are predicted as QTLs that contribute to the attribute of interest [85]. These approaches from pairwise crosses have been successfully applied to map yeast genetic variation responsible for nitrogen utilization [86], metabolic fluxes, ethanol tolerance [87], and high-temperature fermentation [88].

Most crosses constructed in yeast have involved the reference laboratory strain S288c or its derivatives crossed against a wild or fermentative isolate [89]. However, these strains only harbor a small fraction of the phenotypic variation of natural populations and have mosaic genomes of the founder strains [84,90]. Therefore, studies using biparental crosses provide a poor understanding of the relationship between the genetic background and the QTLs. These studies lack resolution since few generations are used; consequently, they are unable to reveal the complete architecture of polygenic traits. Moreover, laboratory strains often contain artificial auxotrophic markers that confound mapping experiments [91]. Investigators have recently established advanced-generation multi-parent populations (MPPs) in yeast to overcome these problems (Figure 3).

### 4.2. Multi-Parent Populations (MPPs)

Yeast MPPs comprise large populations with thousands to millions of individuals obtained from two main steps. Firstly, several (inbred or isogenic) founder strains from various geographical origins are crossed, and then the intercross of the resulting population is subsequentially crossed for several generations [81]. Large segregating populations are then used for mapping QTLs. The first MPP in yeast was established by Cubillos et al. [92] by crossing four strains representative of the main *S. cerevisiae* lineages (Y12 strain as representative of the SA lineage, YPS128 of the NA lineage, DBVPG6044 of the WA, and DBVPG6765 of the WE lineage) for 12 generations. The SGRP-4X contains 165 sequenced segregants, representing recombined genetic mosaics of the founder strains. Later, Linder et al. [93] extended this approach and created 18F12v1 and 18F12v2, two outbred MPPs derived from a cross of 18 genetically diverse founder strains, with each strain derived from the SGRP collection [84,92,93].

MPPs in yeast are robust mapping resources due to multiple founders and rounds of recombination in many individuals that increase both the genetic and the phenotypic diversity, s well as the linkage block resolution of the QTL mapping compared to biparental F1 or F2 populations. In fact, in yeast, it has been shown that only a few rounds of meiosis are sufficient to obtain spaced near-genic resolution [94]. Association mapping in MPPs provides more equilibrated allelic frequencies than biparental populations, increasing knowledge about the population structure [95]. Integration of this information in the QTL analysis can reduce the probability of obtaining false-positive results, thus demonstrating yeast as an accurate model system to identify dozens to hundreds of genes underlying phenotypes of interest.

### 4.3. Genome-Wide Association Studies (GWAS) in S. cerevisiae

GWAS utilizes the variation in large populations of unrelated individuals to provide insights into the causes of common complex traits. However, in 2012, only 36 *S. cerevisiae* genomes were available from the Saccharomyces Genome Resequencing Project, hampering GWAS studies in yeast. This situation motivated the development of a project to describe whole-genome sequence variation in numerous yeast populations (http://1002genomes.u-strasbg.fr/ (accessed on 22 May 2022)). Today, more than 2000 genomes isolated from a wide range of locations (including Australia, Europe, Russia, Vietnam, and South Africa) are available [96]. Thus, investigators can conduct GWAS in this model organism [97].

The success of GWAS in *S. cerevisiae* is a result of high diversity among natural isolates relative to humans [96], low linkage disequilibrium (extended in an average half-life of <3 kb) [98], and relatively simple quantification of phenotypes in hundreds to thousands of individuals. However, GWAS in yeast is affected by a large population structure [84,98], leading to limited statistical power and spurious associations. The increment in the number of genotyped individuals is comparable to other model organisms enabling GWAS to describe copy number variants (CNV) as having a more significant phenotypic effect than SNV in yeast and laying the foundation for GWAS in the species [99].

Many of the phenotypes addressed in yeast are directly related to the cell-autonomous features of human diseases, including neurological conditions such as Parkinson’s disease [100]. Thus far, most of the disease genome-wide screenings in *S. cerevisiae* have deleted one gene at a time. To our knowledge, the genomic variability of yeast isolates is starting to be used for modeling human phenotypic variabilities. In the field of longevity and environment, a study in which 58 natural yeast strains were used led to identifying *RIM15* and *SER1* as longevity genes under caloric restrictions [101].

In the future, we expect to observe increased research using panels of organisms, where a combination of variants can be identified. This technique could be feasible in the short term for diseases that can be mimicked pharmacologically and in the medium term for disorders that can be reproduced genetically.

## 5. Practical Considerations and Concluding Remarks

Each of the discussed organisms and panels has advantages and disadvantages for human translational research. In addition to the already mentioned ones, researchers should consider practical factors for deciding the best model for each project. Some relevant factors are presented in Table 1.

In conclusion, the consequences of a genetic mutation can be strongly modified by the biological background in which it operates. For example, a loss-of-function mutation may be well tolerated in one genetic context and lethal in another. The most resistant individuals have the biological secrets useful for developing therapies for the most susceptible ones. Human studies are challenging; they can take a long time due to the recruitment of large cohorts, and genomic sequencing is expensive. Instead, modeling diseases in already sequenced panels of diverse model organisms followed by gene mapping and validation in smaller human cohorts can speed up translational research and precision medicine for both common and rare diseases.

## Figures and Tables

**Figure 1 ijms-23-07570-f001:**
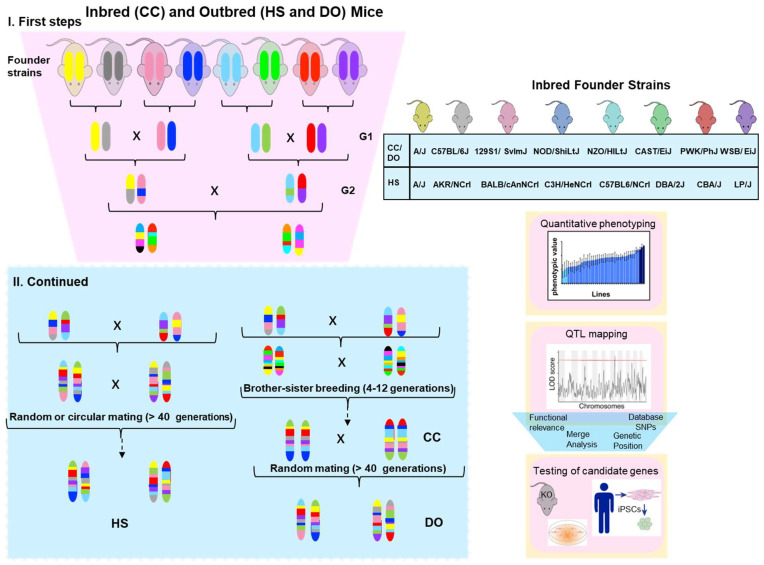
Breeding schemes for inbred (CC) and outbred (HS and DO) mice populations: Inbred founder strains for each panel are indicated in the right box. CC and DO populations share the same eight founder strains, five of which are standard laboratory inbred strains, while three are wild-derived strains. Colors represent the genotypes of strain chromosomes. The first steps include the combination of all eight founder genomes (outcrosses). CC is then generated as a recombinant inbred (RI) after multiple brother–sister breeding. HS and DO panels were developed as high-diversity outbred panels by over 40 generations of random outcrosses. DO was created from partially inbred Collaborative Cross (CC) mice. Quantitative phenotyping can be performed in the strains and used for gene mapping. Some signals in chromosomal locations will probably pass the threshold of significance (red line) in the LOD plot. The functional relevance of these variants can be assessed in animal models such as knockout mice and induced pluripotent stem cells (iPSC) derived from patients.

**Figure 2 ijms-23-07570-f002:**
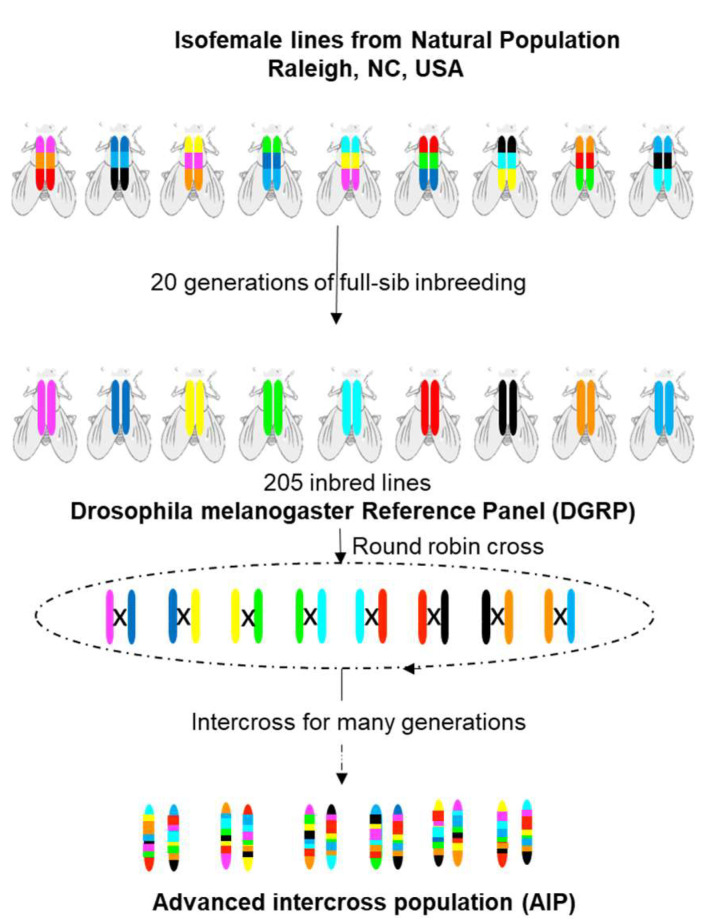
Generation of *Drosophila melanogaster* Genetic Reference Panel (DGRP) and Advanced Intercross Population (DGRP-AIPs). The DGRP corresponds to a sequenced panel derived from a natural fly population of Raleigh, NC (USA), and it was generated through 20 generations of full-sibling mating. The AIPs lines were derived from the DGRP by round-robin crossing and were then remapped.

**Figure 3 ijms-23-07570-f003:**
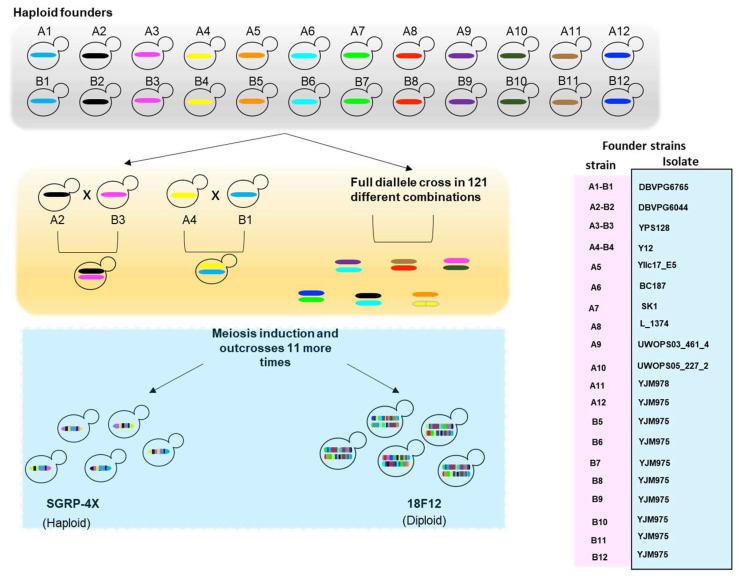
Cross design of SGRP-4X and 18F12 mapping populations. Haploid founder strains used for generations of these populations are indicated in the right box. Ax and Bx indicate the Mat a and Mat α haploid founder strains, respectively.

**Table 1 ijms-23-07570-t001:** Practical considerations for choosing model organisms and their panels. The references are shown in brackets. When deciding the best model for a project, variables such as the percentage of homolog genes to human disease-causing genes, costs, and the possibility of automatization should be considered.

	*Mus musculus*	*Drosophila melanogaster*	*Saccharomyces cerevisiae*
Genome size (kb)	2,725,521 [102]	180,000 [103]	12,070 [104]
Percentage of homolog genes to human disease-causing genes	99 [105]	70 [47,106]	60 [107]
Costs to keep the panels	High	Medium	Very low
Complex behaviors	Yes	Yes	No
Discovery of cell-autonomous processes	Yes	Yes	Yes
Speed for throughput screenings and automatization of measurements	Slow	Fast	Very fast

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
