# Peer review of "Genetic Background Matters: Population-Based Studies in Model Organisms for Translational Research"

_ijms, 2022, doi:10.3390/ijms23147570_

Round 1

Reviewer 1 Report

This could be an interesting area to review.  Unfortunately this manuscript is too flawed to publish as is.

The intro and conclusions should be flushed out more, since they emphasize the use of model organism panels instead of single strains.

The writing is very hard to follow, the grammar and English word usage is just not correct, making it difficult to follow the arguments.  Compound sentences and lists should be taken apart and rewritten to make sense.  They can then be combined for emphasis or variety, but the reader must be able to understand the meaning. 

Below are listed only a few of the stumbling blocks found upon first reading.

The examples of each model that has been applied with positive outcome is interesting but very vague, was it the intention of these authors to send the reader away from this manuscript in order to understand it?

Line 47   strain that are  enabling the study

Line 48 tissues that are available

Line 55.  All abbreviations should be written out on FIRST use

Line 68  composed of (why is comprised so maligned??)

Line 71  only a mouse geneticist would understand this alphabet soup

Line 75 Nucleotide?

Line 81 access to or availability of

Line 86 as for many different, as for any model???

Line 126 a list of one ??

Every list of benefits, considerations etc, rapidly breaks down into a divergent discussion of one aspect or another.  There is no way to follow a complete list. Please separate the lists from the discussion of each component  of the list

Line 149 associated how?  severity? Resistance??

Line 158 matings not mattings

Line 228  does one need All the genetic variation available in a species for informative studies???  Are the mouse panels completely representative of variation in that species???

Line 267  same reference 3x??

Line 278  occur?? Instead of accur?

Line 309 no idea what they are trying to say dreading masking effect less like

Line 33 one   why are the sentence in this argument numbered??

Line 359 resykt if ??? result of//

Author Response

Dear editor and reviewers

First of all, I would like to apologize. I confused the manuscript versions and submitted an unedited draft with too many typos and grammatical mistakes. In this new version, we corrected them, keeping the manuscript's ideas. We did not change the bibliography. You can find two versions of the manuscript, one with track changes and a clear version as well. For this reason, I believe it is not necessary to respond that we modified accordingly each of the absolutely relevant points highlighted by the reviewers.

Thanking you in advance for understanding my mistake,

Sincerely,

Andrés D. Klein, Ph.D.

Reviewer 2 Report

Review report for “Genetic background matters: Population-based studies in model organisms for translational research” by Olguín et al. 

This is a review manuscript describing resources of experimental animal cohorts with variable genetic background for the models of genetic diversity of human clinical study.  I do not have major concerns of the contents of this manuscript, but I found a few typos and issues on formatting as describe below. 

1. Abbreviations: Please revise the manuscript thoroughly, and make sure that the all abbreviations are written out in full when first mentioned. Examples are CC (line 55), DO (line 56), HS (line 56), RI (line 63), MAF (line 231), PD (line 252), MPPS (line 346), etc... 

2. Probable typos: Examples are “sincrease” (line 162: “increases”?), “enabling Identification” (line 174: “enabling identification”?), “Mapping Precision” (line 183: “Mapping precision”?), “accur” (line 278: “occur”?), “arenot” (line 326: “are not”), “MultiParent Populations” (line 341: “Multi Parent Populations”?), “resykt” (lien 359: “result”?).

3. Typos in citation?: “(51),(51),(51)” (line 267), “(76), S(76), (76)” (line 305).  

4. The following sentence was not very much plain for me:  

“Mapping Precision with high-resolution achieved with the use of HS or DO mice has an inverse relationship with statistical power [43], and because a large number of animals are needed for sufficient statistical power, which is not always possible.” (line 183-) 

Perhaps, “Mapping precision with high-resolution achieved with the use of HS or DO mice has an inverse relationship with statistical power [43]. In addition, a large number of animals are needed for sufficient statistical power, but it is not always possible.”? 

5. The following sentence was also not plain for me: 

“One of these research outcomes highlights the ability of knockouts of closely related gene products' dreading masking effect to be less like in yeast than in other complex systems, given the reduced genetic redundancy in this model organism [79].” (line 307-)

Author Response

(The authors gave the same response as above.)
